# Effects of Chronic Kidney Disease on Nanomechanics of the Endothelial Glycocalyx Are Mediated by the Mineralocorticoid Receptor

**DOI:** 10.3390/ijms231810659

**Published:** 2022-09-13

**Authors:** Benedikt Fels, Arne Beyer, Violeta Cazaña-Pérez, Teresa Giraldez, Juan F. Navarro-González, Diego Alvarez de la Rosa, Franz Schaefer, Aysun K. Bayazit, Łukasz Obrycki, Bruno Ranchin, Johannes Holle, Uwe Querfeld, Kristina Kusche-Vihrog

**Affiliations:** 1Institute of Physiology, University of Luebeck, D23562 Luebeck, Germany; 2DZHK (German Research Centre for Cardiovascular Research), Partner Site Hamburg/Luebeck/Kiel, D23562 Luebeck, Germany; 3Institute of Physiology II, University of Muenster, D48149 Muenster, Germany; 4Department of Basic Medical Sciences, Instituto de Tecnologías Biomédicas, Universidad de La Laguna, ES38206 La Laguna, Spain; 5Unidad de Investigación and Servicio de Nefrología, Hospital Universitario Nuestra Señora de Candelaria, ES38010 Santa Cruz de Tenerife, Spain; 6Pediatric Nephrology Division, Center for Pediatrics and Adolescent Medicine, University of Heidelberg, D69120 Heidelberg, Germany; 7Department of Pediatric Nephrology, Cukurova University, Adana TR01250, Turkey; 8Department of Nephrology, Kidney Transplantation and Hypertension, Children’s Memorial Health Institute, PL04-730 Warsaw, Poland; 9Pediatric Nephrology Unit, Hôpital Femme Mère Enfant, Hospices Civils de Lyon, Université de Lyon, F69002 Lyon, France; 10Department of Pediatrics, Division of Gastroenterology, Nephrology and Metabolic Medicine, Charité—Universitätsmedizin Berlin, Corporate Member of Freie Universität Berlin and Humboldt-Universität zu Berlin, D13353 Berlin, Germany; 11DZHK (German Research Centre for Cardiovascular Research), Partner Site Berlin, D10785 Berlin, Germany

**Keywords:** endothelium, glycocalyx, mineralocorticoid receptor, ENaC, chronic kidney disease

## Abstract

Endothelial mechanics control vascular reactivity and are regulated by the mineralocorticoid receptor (MR) and its downstream target, the epithelial Na^+^ channel (ENaC). Endothelial dysfunction is a hallmark of chronic kidney disease (CKD), but its mechanisms are poorly understood. We hypothesized that CKD disrupts endothelial mechanics in an MR/ENaC-dependent process. Methods: Primary human endothelial cells were cultured with uremic serum derived from children with stage 3–5 (predialysis) CKD or adult hemodialysis (HD) patients or healthy controls. The height and stiffness of the endothelial glycocalyx (eGC) and cortex were monitored by atomic force microscopy (AFM) using an ultrasensitive mechanical nanosensor. Results: In a stage-dependent manner, sera from children with CKD induced a significant increase in eGC and cortex stiffness and an incremental reduction of the eGC height. AFM measurements were significantly associated with individual pulse wave velocity and serum concentrations of gut-derived uremic toxins. Serum from HD patients increased MR expression and mechanical stiffness of the endothelial cortex, an effect reversed by MR and ENaC antagonists, decreased eNOS expression and NO bioavailability, and augmented monocyte adhesion. Conclusion: These data indicate progressive structural damage of the endothelial surface with diminishing kidney function and identify the MR as a mediator of CKD-induced endothelial dysfunction.

## 1. Introduction

Patients with chronic kidney disease (CKD) have a markedly increased incidence of cardiovascular events and mortality compared to the age-matched general population. Endothelial dysfunction is recognized as a principal disturbance of the vascular tree occurring in patients with CKD, even at early stages of diminished kidney function [1]. The endothelium in CKD is characterized by a change from a quiescent to an activated phenotype, resulting in impaired vasodilatation and a proinflammatory and prothrombotic state [2]. However, mechanistic knowledge of CKD-induced mechanical changes in endothelial cells is limited.

Endothelial cells form a continuous layer of the entire vascular tree, providing a biologic interface between blood and vessels, which is highly responsive to changing requirements by dynamic alterations of the mechanical properties (i.e., stiffness). The luminal surface of the endothelium is covered by the negatively charged, brush-like endothelial glycocalyx (eGC), a multifunctional layer of membrane-bound, carbohydrate-rich molecules, mostly consisting of glycoproteins and proteoglycans [3,4,5]. Together with the underlying cortex, an actin-rich layer 50 to 150 nm beneath the plasma membrane, the eGC has the ability to respond to mechanical stress, resulting in an adapted bioavailability of vasoactive substances such as nitric oxide (NO) [3,6]. Surface properties of endothelial cells are mediated by the mineralocorticoid receptor (MR) and its downstream target, the amiloride-sensitive epithelial Na^+^ channel (ENaC) [7,8]. Pharmacological as well as genetic manipulation of ENaC function or expression alters cortical stiffness [9]. A reduced ENaC membrane abundance and/or activity, in turn, increases the endothelial NO release, mechanisms linking endothelial cell mechanics with endothelial function [10,11].

Previous studies have provided evidence for CKD-induced injury of the eGC. Thus, an incremental increase in the proteoglycans syndecan-1 and hyaluronan has been found in the serum of CKD patients, suggesting the shedding of these eGC components due to endothelial damage [12]. The endothelial surface layer of the sublingual microvasculature—a surrogate non-invasive measurement of the eGC height in patients—was found to progressively decrease dependent on kidney function [13], and one recent study found a significant association with serum levels of gut-derived uremic toxins [14]. High concentrations of uremic toxins have many deleterious effects on endothelial cells, inducing a proinflammatory, prothrombotic state and endothelial dysfunction [15].

Here, we hypothesized that serum obtained from CKD patients with either predialysis CKD (stage 3–5) or on maintenance hemodialysis (“uremic serum”) directly affects the mechanical properties of the endothelial cell surface, contributing to endothelial dysfunction and vascular inflammation. Further, we examined the roles of MR and its downstream target, ENaC, in the process.

## 2. Results

### 2.1. The eGC Height Decreases Stage-Dependently in Children with CKD

To investigate whether the eGC is affected by CKD, HUVECs were incubated with serum from pediatric patients with CKD 3–5 for 24 h, and eGC height and stiffness were measured by AFM nanoindentation. The eGC height was decreased by 21 ± 0.01%, 24 ± 0.01% and 30 ± 0.01%, respectively (N ≥ 7; **** *p* < 0.0001 control vs. CKD), suggesting a deleterious effect of CKD, depending on severity, on the eGC structure (Figure 1A). This effect is visualized by eGC fluorescence staining using WGA (Figure 1B). HUVECs incubated with 10% pooled uremic serum from pediatric pre-dialysis patients showed, although not incrementally, significant decrease in eGC fluorescence intensity at all stages. The fluorescence intensity of eGC/WGA was diminished by 23.5 ± 2.3%, 17.8 ± 2.3% and 9.3 ± 3.1% (Figure 1C), confirming structural damage to the eGC in CKD patients (N = 7–25; n = 76–170, ** *p* < 0.01, *** *p* < 0.001, **** *p* < 0.0001). Corresponding to these findings, eGC stiffness was gradually increased up to 7 ± 0.02% in CKD stage 5 (Figure 1D, N = 7–24, ** *p* < 0.01 control vs. CKD 5).

### 2.2. Uremic Serum Induces Cortical Stiffness and Increases Actin Content in Endothelial Cells

We then studied the cortical stiffness of endothelial cells. As shown in Figure 1E, incubation of HUVECs with 10% serum from pediatric patients for 24 h led to an incremental increase in cortical stiffness by 20.1 ± 0.01%, 22.5 ± 0.01% and 37.5 ± 0.01% with each stage of CKD compared to control (N = 7–24, n = 182–612; * *p* < 0.05, **** *p* < 0.0001), indicating progressive development of endothelial dysfunction.

To further analyze structural changes and cortical stiffness, HUVECs were incubated with 10% serum from adult hemodialysis patients for 24 h. Treatment with uremic serum derived from adult patients increased cortical stiffness significantly by 23.6 ± 2.0% compared to controls (Figure 2A; N = 6; n = 48–55; **** *p* < 0.0001 control vs. uremia). Stiffening of the actin cortex was confirmed by FITC-labeled phalloidin fluorescence staining of F-actin fibers (Figure 2B). Quantification of phalloidin fluorescence intensity showed an increase in actin polymerization/fluorescence intensity of 22.1 ± 4.4% compared to controls (Figure 2C, N = 3, n = 80, *** *p* < 0.001 control vs. uremia).

### 2.3. Uremic Serum Increases Cortical Stiffness in an MR-Dependent Fashion

Since aldosterone signaling through the MR has been implicated in cortical stiffening of the endothelium, we studied the effect of uremic serum on the MR. HAEC were treated with medium supplemented with 20% control or uremic serum in the presence or absence of 0.1 µM spironolactone (MR antagonist) for 5 days. As shown in Figure 2D, MR gene expression is induced by uremic serum and reversed by treatment with spironolactone (N = 3; * *p* < 0.05 control vs. uremic serum). Treatment of HUVECs with uremic serum derived from adult patients increased the ENaC membrane abundance by +29.4 ± 0.1% compared to control (Figure 2E), which was effectively prevented by spironolactone, but not amiloride (uremia +27.3 ± 0.1% *** *p* < 0.001, uremia+Spiro −6.7 ± 0.1% ns *p* > 0.05, all vs. control serum).

Furthermore, inhibition of MR with spironolactone slightly decreased cortical stiffness in control cells (Figure 2F control 1.23 ± 0.01 pN/nm vs. control+Spiro 1.16 ± 0.02 pN/nm), but completely abolished the effect of uremic serum (uremia 1.45 ± 0.03 pN/nm vs. uremia+Spiro 1.17 ± 0.02 pN/nm; N ≥ 7; * *p* < 0.05, **** *p* < 0.0001). While functional inhibition of ENaC with 1 µM amiloride led to decreased cortical stiffness by −16.0 ± 1.6% in control cells (Figure 2G) in agreement with previous results [9], the amiloride effect was abolished in HUVECs treated with uremic serum. These data indicate that uremic serum significantly increases the cortical stiffness of endothelial cells, and that this effect is mediated by MR signaling and ENaC.

Since monocyte adhesion is one of the first steps in the pathogenesis of vascular inflammation, we tested the impact of uremic sera on the adhesion of human monocytes to vascular endothelial cells. Treatment with 10% uremic sera (24 h) resulted in a threefold increase of adherent monocytes compared to control conditions (Figure 2H) (#monocyte per cell in control: 0.133 ± 0.009 vs. uremia: 0.419 ± 0.022, **** *p* < 0.0001).

### 2.4. Uremic Serum Decreases eNOS Protein Abundance and NO Bioavailability

Diminished height of the eGC and increased cortical stiffness suggest that uremic serum induces endothelial dysfunction in cultured endothelial cells. This was confirmed by experiments showing that uremic serum decreases eNOS protein abundance and NO bioavailability in HAEC (Figure 3). Nitric oxide production was significantly diminished by incubation with uremic serum from adult dialysis patients, which was not prevented by treatment with spironolactone (Figure 3A). In contrast, *eNOS* mRNA expression was not affected by uremic serum or spironolactone (Figure 3B). Compared to control serum, treatment with uremic serum resulted in a decreased relative eNOS protein concentration, which was prevented by spironolactone (Figure 3C,D). Thus, while *eNOS* gene expression was unchanged, eNOS protein expression and NO bioavailability were significantly diminished by uremic serum, and only the latter effect was unchanged by MR inhibition with spironolactone. These results indicate that diminished endothelial NO production (endothelial dysfunction) cannot be prevented by MR inhibition alone but is likely mediated by additional mechanisms.

### 2.5. Correlation with Clinical Data of Children with CKD

To explore potential associations between the *in-vitro* data with individual clinical characteristics of pediatric patients (4C study), we calculated linear correlations of each patient’s clinical and selected biochemical data with the impact of their sera on eGC properties.

The eGC height correlated negatively with the CKD stage (r = −0.67, *p* < 0.01), whereas eGC stiffness (r = 0.30, *p* < 0.07) and cortex stiffness (r = 0.43, *p* < 0.02) correlated positively with the CKD stage (see Table 1). The height of eGC showed a significant inverse correlation with indoxyl-sulfate (IS; r = −0.37, *p* = 0.04) as well as p-cresyl-sulfate (PCS; r = −0.50, *p* = 0.01) serum concentrations. In addition, eGC stiffness was correlated with PCS concentrations (r = 0.35, *p* = 0.05) and cortex stiffness was positively correlated with Angpt-2 (Angiopoietin-2), a marker for endothelial dysfunction (r = 0.82, *p* < 0.01). Both eGC (r = 0.37, *p* = 0.03) and cortex stiffness (r = 0.37, *p* = 0.04) showed positive correlations with the pulse wave velocity (PWV) as a surrogate marker for cardiovascular disease.

## 3. Discussion

The endothelial surface is highly vulnerable to toxins and factors released during acute or chronic diseases, resulting in a loss of vasoprotective function. In this study, the effects of uremic serum derived from CKD patients on the endothelial surface mechanics and function are studied for the first time. We found an incremental decrease in eGC height and, in parallel, an incremental increase in cortical stiffness with each stage of CKD, indicating a progressive development of endothelial dysfunction from stage 3 to 5. This change in the mechanical properties of the endothelial surface could be prevented by inhibition of MR and, at least partially, by inhibition of its downstream target ENaC. Treatment with uremic serum furthermore resulted in increased monocyte adhesion, a reliable marker for inflammatory processes. On the molecular level, uremic serum potently increased MR mRNA expression and ENaC membrane abundance.

These results are in line with the concept of a deleterious role of excess aldosterone/MR signaling in CKD [16]. Of note, plasma aldosterone levels are increased in CKD, and aldosterone has been found to increase vascular inflammation and fibrosis [16]. Recent and ongoing clinical trials show cardioprotective and nephroprotective effects of treatment with non-steroidal selective MR antagonists in CKD patients and further underscore the important role of the MR in CKD-related cardiovascular damage [17].

Exposure of primary endothelial cells to uremic serum significantly decreased NO production and eNOS expression, indicating the development of endothelial dysfunction under such conditions. This could not be prevented by the inhibition of MR using spironolactone.

CKD is a state of persistent inflammation, and thus the eGC, due to its position on top of endothelial cells, is a vulnerable target for circulating inflammatory mediators such as cytokines, C-reactive protein, alkaline phosphatase and various uremic toxins [15,18]. One example is endotoxin, a gut-derived uremic toxin, of which high circulating levels have been documented in CKD patients [19]. Infusion of low-dose endotoxin in healthy volunteers resulted in a decrease in the sublingual microvascular glycocalyx thickness, which could be attenuated by TNF-alpha inhibition [20].

In the present study, the functional height of eGC was drastically reduced after incubation of primary endothelial cells with sera derived from pediatric CKD patients, a unique population without significant vascular risk factors related to old age, lifestyle, or diabetes. Intriguingly, the eGC height incrementally decreased from CKD stage 3–5. In parallel, the eGC and cortical stiffness increased, indicating progression of endothelial damage and paving the way for inflammatory processes [3].

The effects of uremic serum are most likely conferred by multiple pathways, but our data suggest an important contribution from gut-derived uremic toxins, which accumulate in serum with decreasing kidney function [21]. Protein-bound solutes such as IS and PCS are not inert biomarkers of kidney dysfunction, but have a multitude of specific deleterious effects on endothelial cells [15]. In the present study, we found that IS and PCS serum concentrations were inversely correlated with eGC height. Of note, in the present study, we could also demonstrate augmented monocyte adhesion under uremic conditions. Intravenous infusion of IS resulted in direct eGC injury as evidenced by shedding of heparin sulfate and a reduction in eGC volume [22]. These data are in line with a study in CKD patients showing a correlation of glycocalyx injury (syndecan-1 and hyaluronan levels) and endothelial dysfunction (VCAM-1) with IS and PCS concentrations [14]. Taken together, our data confirm an important role of uremic toxins in eGC injury and endothelial dysfunction.

As mentioned above, uremic serum from pediatric patients also significantly increased cortical stiffness in a CKD stage-dependent manner along with cortical actin content. Since the eGC and the cortex are mechanically and functionally connected [3], these findings indicate reorganization of the endothelial cortex and a profound alteration in the nanomechanical properties of the eGC. Thus, the endothelial surface can be seen in total as a functionally important compartment which is vulnerable to pro-inflammatory factors. Of note, the eGC stiffness as well as the cortical stiffness were significantly associated with the velocity of the pulse wave, which is in line with the concept of increased systemic vascular stiffness in CKD [23] and the concept of crosstalk between the microcirculatory and arterial properties [24,25].

Not surprisingly, uremic serum decreased eNOS protein abundance and NO bioavailability, the hallmark of endothelial dysfunction. The NO bioavailability could not be rescued by treatment with spironolactone, indicating the presence of additional inhibitors/mechanisms suppressing NO release, which are abundant in uremic serum [26], but were not measured in the present study.

The vascular growth factors Angpt-1 and its antagonist Angpt-2 are important physiological regulators of angiogenesis, endothelial cell permeability and survival [27]. Since Angpt-2 was shown to induce vascular leakage in inflammatory conditions, implying interaction with the eGC, we have previously investigated effects on the eGC; we were able to show that Angpt-2 (100 ng/mL) mediates heparinase-dependent breakdown of the eGC thickness while cortex stiffness was unaffected [28]. In the present study, using the same methods, we show a close correlation (r = 0.82) of Angpt-2 with cortical stiffness but no significant correlation with eGC height. However, Angpt-2 serum concentrations are much lower in serum from CKD patients (5–10-fold) than when applied alone with solvent [25]. These findings provide further evidence that Angpt-2 has important deleterious effects on eGC properties and might be a therapeutic target for protection of the vascular barrier in CKD.

The present study has some limitations. The small number of subjects is a clear limitation for the calculation of statistical correlations of vascular properties and clinical characteristics measured in the 4C-study, and the results should be interpreted with caution. Similarly, the limited size of the study does not permit definitive conclusions regarding the specificity of the effects of uremic toxins on eGC properties since other toxins or further noxious factors in serum were not investigated.

We conclude that CKD, in a stage-dependent manner, generates structural and mechanical changes of the endothelial surface, including the eGC and the cortex via the aldosterone–MR axis. Together with a decreased bioavailability of NO, this could be an important mechanism involved in the development of endothelial dysfunction, preceding irreversible damage of the vasculature and eventually resulting in CKD-associated cardiovascular events. Thus, our study confirms the important role of the MR in CKD-related cardiovascular disease. Changes in eGC properties were associated with concentrations of the uremic toxins IS and PCS in serum, identifying targets for new therapeutic approaches.

## 4. Materials and Methods

### 4.1. Cell Isolation and Culture

Primary human umbilical vein endothelial cells (HUVECs) were isolated from umbilical cords donated by patients giving birth in the “Marienkrankenhaus Luebeck” (approved by Local Ethics Committee Case: 18-325), as described before [18]. Cells were cultured in standard cell culture media culture medium (Gibco Medium 199 + Fetal Calf Serum 10% (Gibco, Carlsbad, CA, USA) + Penicillin/Streptomycin 1% (Gibco, Carlsbad, CA, USA) + Large Vessel Endothelia Supplement 1% (Gibco, Carlsbad, CA, USA) + Heparin 5000 U/mL (Biochrom, Schaffhausen, Switzerland). HUVECs were used until passage 3 for experiments. Primary human aortic endothelial cells (HAEC) were obtained from Innoprot (Derio, Spain) and cultured according to the manufacturer’s instructions. When indicated, cells were exposed to control or uremic human sera for five days while treated or not with MR inhibitor spironolactone (1 µM) or ENaC inhibitor amiloride (10 µM). HAECs were used until passage 5 for experiments.

### 4.2. Human Uremic Serum

Uremic sera derived from pre-dialytic children with CKD were obtained from patients participating in the Cardiovascular Comorbidity in Children with Chronic Kidney Disease (4-C) Study (see Appendix B Table A1) [29]. This prospective observational cohort study, conducted at 55 pediatric nephrology centers in 12 European countries, was approved by the Ethics Board of the University of Heidelberg (S-032/2009) and subsequently by the local review boards of each participating institution in accordance with the Declaration of Helsinki. The 4-C study included children with an initial eGFR of 10–60 mL/min per 1.73 m^2^, i.e., stage 3–5 CKD, and sera obtained from 25 patients were used. The estimated glomerular filtration rate (mL/min/1.73 m^2^) of these patients [30] was 41.3 in patients (n = 10) with stage 3, 17.5 in patients (n = 8) with stage 4, and 12.6 in patients (n = 7) with stage 5 CKD (all pre-dialysis). All data achieved by stimulation with sera from pre-dialytic children with CKD are indicated as “CKD3-5” and plotted in Figure 1.

As surrogate markers of CVD in children, the carotid intima media thickness (cIMT) and the pulse wave velocity (PWV) were measured as described, and age-, gender and height-adjusted standard deviation scores (SDS) were used for calculations [29]. Serum biomarkers Angiopoietin-1 (Angpt-1) and Angiopoietin-2 (Angpt-2) and the uremic toxins indoxylsulfate (IS) and p-cresylsulfate (PCS) were measured as part of the 4C-study [21] (see Appendix A).

In a second series of experiments, healthy adult donor and patient sera were obtained at the Nephrology Service, Hospital Universitario Nuestra Señora de Candelaria (Tenerife, Spain) after approval by the Hospital Ethics Committee (Ref. PI-19/13). Human subjects agreed to sample collection in writing and in agreement with the Declaration of Helsinki. Selection criteria and analytical characteristics of the serum samples have been reported elsewhere [31,32]. After individual serum analysis, samples were pooled, aliquoted and stored at −80 °C until use. Throughout this study, we used serum pools from 16 healthy donors and 16 stage 5D CKD patients with an estimated glomerular filtration rate ≤ 15 mL/min/1.73 m^2^ (indicated as “uremia”).

### 4.3. Immunofluorescence Staining

The eGC was stained using Alexa488 conjugated wheat germ agglutinin (WGA, Thermo Fisher; Waltham, MA, USA) staining. After treatment, cells were immediately fixed with 0.1% glutaraldehyde for 30 min and incubated with WGA (1:500) for 1 h at RT. After the washing steps, the cells were mounted with Dako mounting medium (Dako, Carpinteria, CA, USA) containing Hoechst (Sigma Aldrich, St. Louis, MO, USA; 1.5 µg/mL) for nuclei staining. If stainings for visualization of the actin cytoskeleton were done with Phalloidin-TRITC (10 µg/mL, 1 h at RT, Sigma Aldrich, St. Louis, MO, USA) after cell fixation with 3.5% paraformaldehyde for 30 min on ice. Cells were permeabilized for 10 min with 0.1% Triton X-100 (Sigma Aldrich, St. Louis, MO, USA), and blocked with 10% normal goat serum for 30 min. After washing, the cells were mounted as described above. ENaC staining was done on PFA-fixed cells without permeabilization. After blocking with 10% normal goat serum for 30 min, coverslips were incubated with rabbit anti-human α-ENaC polyclonal antibody (1:200, Bioss Antibodies Inc., Woburn, MA, USA) and stained with anti-rabbit secondary antibody conjugated to Alexa 488 always for 1h, RT. After washing, the cells were mounted with Dako mounting medium. Fluorescence images were captured with an inverted confocal microscope (TCS SP8, Leica Microsystems, Wetzlar, Germany) equipped with a 63× NA 1.4 objective, or with a Keyence Fluorescence Microscope BZ9000 (Keyence Corporation, Osaka, Japan; 60× objective). The fluorescence intensity of WGA and Phalloidin-TRITC-staining were analyzed by using ImageJ software (Version 1.52a; National Institute of Health, Bethesda, MD, USA). Analysis of the ENaC channel abundance was done with the YT-Evaluation software (Version 2.1.12014; Synentec, Elmshorn, Germany).

### 4.4. Quantification of Gene Expression

Total RNA was purified from cultured HAEC using a commercially available kit (Total RNA Spin Plus Kit, REAL, Valencia, Spain) that includes in-column DNase I digestion to prevent carryover of genomic DNA. The total RNA concentration was estimated by absorption spectrophotometry. Expression of endothelial dysfunction markers and eNOS expression was performed by qPCR using TaqMan^®^ probes. Relative quantification of mRNA abundance was performed using the ∆∆Ct method [33] using GAPDH as an internal reference.

### 4.5. Western Blot

HAEC were lysed in TENT buffer (in mM: 50 Tris-HCl, 5 EDTA, 150 NaCl, 1% Triton X-100, pH 7.4) containing protease inhibitors (Complete^TM^, Roche, Basel, Switzerland). Centrifugation supernatants were quantified using the bicinchoninic acid method (BCA, Sigma). Equal amounts of total protein were mixed with Laemmli buffer, resolved on SDS-PAGE and transferred to polyvinylidene difluoride (PVDF) membranes using the Transblot Turbo transfer pack (Bio-Rad, Hercules, CA, USA). Total eNOS abundance was detected using a rabbit polyclonal antibody (Cell Signaling, Danvers, MA, USA, catalog number 9572; RRID: AB_329863) at a 1:1000 dilution. Mouse anti-GAPDH monoclonal antibody was used as protein loading control (Abcam, Cambridge, UK, catalog number ab9484; RRID: AB_307274) at 1:3000. Secondary antibodies were anti-mouse or anti-rabbit IgG conjugated to horseradish peroxidase (GE Healthcare, Chicago, IL, USA, catalog #NA931 and #934; RRIDs: AB_772210 and AB_772206) and were both used at 1:20,000. Signals were developed using Clarity^TM^ Western ECL substrate (Bio-Rad), detected with a Chemidoc imaging system (Bio-Rad) and quantified with Quantity One^®^ software (version 4.6.7, Bio-Rad).

### 4.6. Quantification of NO Production

The formation of NO was assessed by measuring the accumulation of the stable breakdown product nitrite in the supernatant of HAEC and HUVEC with the Griess reagent [34] as previously described [35]. Briefly, the cell culture supernatants were collected and cleared by centrifugation for 10 min at 18,000× *g*. Equal volumes of 0.02% (*w*/*v*) N-(1-naphthyl)-ethylenediamine and 1% (*w*/*v*) sulfanilamide prepared in 1% HCl were then added to each sample. The mix was incubated for 15 min at RT and the reaction product was quantified by absorption spectrophotometry at 540 nm. Each sample was assayed in triplicate. A standard curve with freshly prepared nitrite standards (eight standards ranging from 0 to 20 µM) was run in parallel with each assay.

### 4.7. Monocyte Adhesion Assay

To identify the effects of uremia on vascular inflammation, the monocyte adhesion on ECs was quantified. Isolation of human monocytes was done by using the pluriSelect monocyte isolation kit (anti-hu CD14, pluriSelect LifeScience, Leipzig, Germany). This kit uses the principle of positive selection from blood samples of healthy donors (approved by the Local Ethics Committee Case: 19-310). Isolated and fluorescently labeled CD14 positive monocytes were then used for a classical wash assay, as described before [36]. Following 24 h of treatment with 10% control or uremic human sera, HUVECs were incubated with isolated monocytes (fluorescently labeled with an Alexa Fluor 488 anti-human CD14 antibody, Biolegend, San Diego, CA, USA). After incubation, non-adherent monocyte cells were removed by washing steps. HUVECs and adherent monocytes were fixed (4% PFA) and quantified by fluorescence microscopy.

### 4.8. Atomic Force Microscopy Measurements

The mechanical stiffness and thickness of the endothelial glycocalyx and cortical stiffness were determined using an AFM nanoindentation technique (MultiMode^®^ SPM, Bruker, Karlsruhe, Germany), with a soft cantilever (spring constant of 10 pN/nm (for eGC) or 30 pN/nm (for cortex); Novascan, Ames, IA, USA) and a polystyrene sphere as the tip (diameter: 10 μm) as described before [37,38]. A maximal loading force of 0.5 nN (for eGC) and 3.0 nN (for cortex) was applied. HUVECs were analyzed in HEPES-buffer (HEPES: 4-(2-hydroxyethyl)-1-piperazineethanesulfonic acid; Buffer (in mM): 140 NaCl, 5 KCl, 1 CaCl_2_, 1 MgCl_2_, 5 glucose, 10 HEPES) supplemented with 1% FCS at 37 °C in a fluid chamber. Stiffness and thickness values were calculated from force-distance curves with the Protein Unfolding and Nano-Indentation Analysis Software PUNIAS3D version 1.0 release 2.2 (Philippe Carl & Paul Dalhaimer, punias@free.fr).

### 4.9. Statistical Analysis

Statistical analysis was performed using GraphPad Prism 6. Data are presented as average ± SE. Comparison between two groups was performed using Student’s *t* test or Mann–Whitney test. Comparison between three or more groups was performed using one-way analysis of variance (ANOVA) followed by Tukey’s post-hoc test. Pearson correlation with one-tailed p values was used to assess associations between CKD stage, CVD biomarkers Angpt-1/2, the uremic toxins IS and PCS as well as cardiovascular surrogate parameters cIMT and PWV (see Appendix A) with the composition of the eGC and actin cortex. Statistical significance was set at *p* < 0.05.

## Figures and Tables

**Figure 1 ijms-23-10659-f001:**
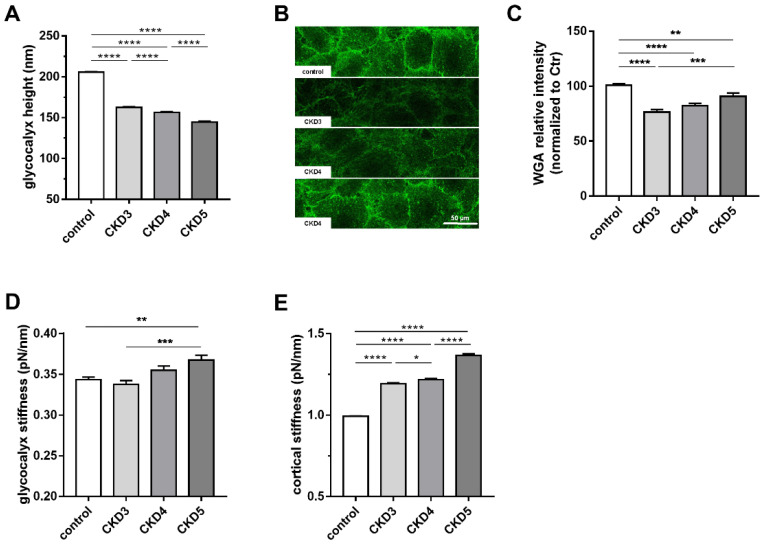
**Incremental effect of diminished kidney function on the structure and mechanics of endothelial cells.** HUVECs were treated with 10% serum from pediatric patients with CKD 3–5 for 24 h. (**A**) eGC height was measured using AFM based nano-intendation measurements. The eGC height was gradually decreased with each stage of CKD (N ≥ 7). (**B**) The decrease in eGC was confirmed by staining with FITC-labeled WGA. (**C**) Treatment with CKD 3–5 sera decreases the WGA fluorescence intensity at all CKD stages compared to controls (N = 7–25; n = 76–170). (**D**) eGC stiffness increases with higher CKD stages (N ≥ 7). (**E**) Stiffness of the actin cortex was measured by nano-intendation measurements. Cortical stiffness increases gradually with each CKD stage (N = 7–24, n = 182–612). Significance levels in all plots are indicated by * *p* < 0.05, ** *p* < 0.01, *** *p* < 0.001, **** *p* < 0.0001.

**Figure 2 ijms-23-10659-f002:**
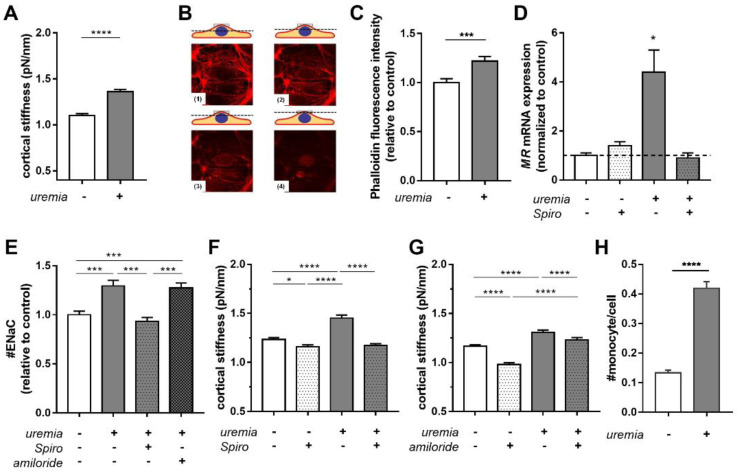
**Uremic serum increases cortical stiffness through the MR/ENaC axis**. Stiffness of cortical actin was measured by using the AFM nano-indentation technique. (**A**) Treatment with 10% uremic serum for 24 h increases the cortical stiffness in HUVECs compared to control-treated cells (N = 6; n = 48–55). (**B**) For validation, TRITC-labeled phalloidin staining was used to quantify cortical actin. Confocal microscopy focuses on the cortical actin layer directly underneath the plasma membrane (see image #4). (**C**) Quantification of phalloidin staining revealed increased actin polymerization in HUVECs treated with uremic serum compared to the control group (N = 3, n = 80). (**D**) HAEC were treated with medium supplemented with 20% of control or uremic serum in the presence or absence of 0.1 µM spironolactone (Spiro) for 5 days. MR mRNA abundance was assessed by qPCR (N = 3). (**E**) Spironolactone prevents uremia induced ENaC membrane insertion (N = 5, n = 142–159). (**F**) Uremic serum leads to increased cortical stiffness, whereas this was not the case in the co-treatment with spironolactone (N = 6, n = 47–75). (**G**) Inhibition of ENaC by amiloride reduces cortical stiffness within the control group. Amiloride also attenuates the effect of uremic serum on the cortical stiffness (N = 4, n = 30–41). (**H**) Monocyte adhesion increases after treatment with uremic serum (N = 4, n = 227–231). Significance levels in all plots are indicated by * *p* < 0.05, *** *p* < 0.001, **** *p* < 0.0001.

**Figure 3 ijms-23-10659-f003:**
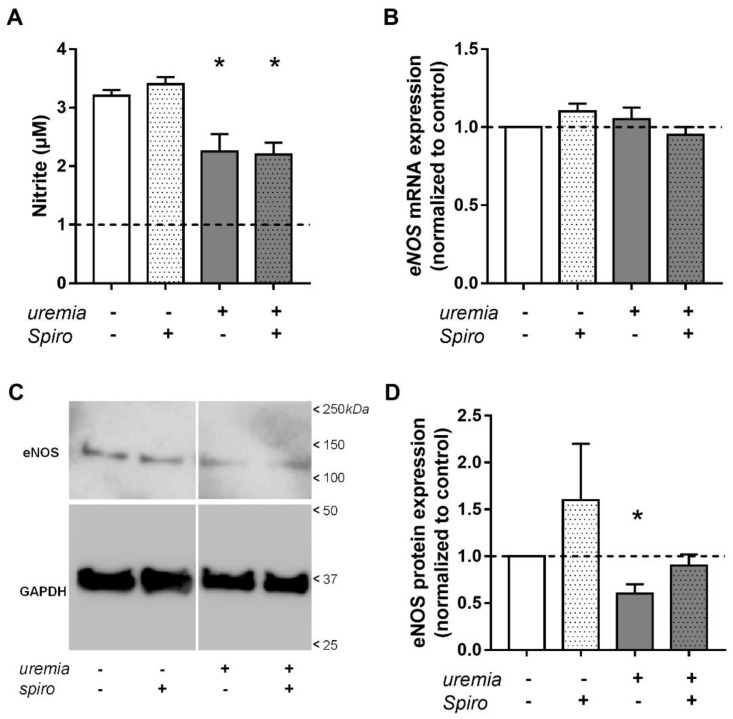
**Uremic serum decreases NO bioavailability and eNOS protein abundance.** (**A**) Nitrite accumulation in HAEC cultured for five days in control (white bars) or uremic serum (grey bars) in the presence (dotted bars) or absence of 0.1 µM spironolactone (Spiro). Bars represent average ± SE (N = 3, *, *p* < 0.05 vs. control). (**B**) *eNOS* mRNA abundance was analyzed by qPCR. Bars represent average fold-change compared to control ± SE (N = 3, no significant change was detected). (**C**) Representative Western blot images obtained using anti-eNOS or anti-GAPDH antibodies. Arrows indicate the relative of molecular mass markers (values in kDa). (**D**) Relative abundance of eNOS protein in HAEC treated as indicated. Bars represent average fold-change compared to control ± SE (N = 3, *, *p* < 0.05 vs. control).

**Table 1 ijms-23-10659-t001:** **Correlations of glycocalyx properties with clinical variables and serum concentrations of gut-derived uremic toxins.** Correlations of glycocalyx properties with clinical data of children with CKD. Variables showing correlations with statistical significance (**bold**). Abbreviations: SDS = standard deviation score, cIMT = carotid intima-media thickness, PWV = pulse wave velocity, IS = indoxyl-sulfate, PCS = p-cresyl-sulfate, Angpt = Angiopoietin, eGC = endothelial glycocalyx.

	CKD Stage	Log IS	Log PCS	Angpt1	Angpt2	cIMT-SDS	PWV-SDS
eGC Height							
r =	**−0.67**	**−0.37**	**−0.50**	−0.13	−0.06	−0.07	−0.23
*p* =	**<0.01**	**0.04**	**0.01**	0.28	0.40	0.37	0.13
eGC Stiffness							
r =	**0.30**	0.33	**0.35**	0.14	−0.11	0.04	**0.37**
*p* =	**0.07**	0.06	**0.05**	0.27	0.30	0.42	**0.03**
Cortex Stiffness							
r =	**0.43**	−0.01	0.10	0.29	**0.82**	−0.22	**0.37**
*p* =	**0.02**	0.48	0.34	0.10	**<0.01**	0.15	**0.04**

## Data Availability

The data presented in this study are available on request from the corresponding author. The data are not publicly available due to required permission of the 4C Study group.

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
