# Peer review of "Effects of Chronic Kidney Disease on Nanomechanics of the Endothelial Glycocalyx Are Mediated by the Mineralocorticoid Receptor"

_ijms, 2022, doi:10.3390/ijms231810659_

Round 1

Reviewer 1 Report

I think this is very excellent study. However, there are some modifications/clarifications needed.

1. How can the investigators indentify/define uremic human sera, and if the BUN cut-off is arbitrary? Any references? given there is no specific cut-off of BUN for uremia in clinical practice

2. Figure 1 is very difficult to evaluate, especially electron microscopy Figure 1D.

Reviewer 2 Report

This submission concluded that MR as a possible mediator of uremia-induced endothelial dysfunction. This study is well designed, data are sound, results from this study provide new information for clinical management. However, there are several issues need to be improved. A revision is suggested.

1.          Please also study p-NOS in fig 3.

2.          Please discuss the clinical implications of this study.

3.          Please discuss the limitations of this study.

4.          Please briefly address methods in the abstraction.

5.          The method of Nitrite accumulation is missed.

6.          How many passages of cells used in this study. Please address.

Round 2

Reviewer 2 Report

My questions had been well addressed. This submission is acceptable.